# Comparative study on chloroplast genomes of three *Hansenia forbesii* varieties (Apiaceae)

**Chenghao Zhu**[1], **Yuan Jiang**[1], **Yu Bai**[1], **Shengjian Dong**[2], **Sun Zhirong**[1]*

**1** School of Chinese Materia Medica, Beijing University of Chinese Medicine. Beijing, Beijing, China,
**2** College of Applied Technology, Gansu Agricultural University. Lanzhou, Gansu, China

* szrbucm67@163.com

## Abstract

To find the gene hypervariable regions of three varieties of *Hansenia forbesii* H. Boissieu and determine their phylogenetic relationship, the chloroplast (cp) genome of these three varieties were firstly sequencing by the Illumina hiseq platform. In this study, we assembled the complete cp genome sequences of *Hansenia forbesii* LQ (156,954 bp), *H. forbesii* QX (157,181 bp), *H. forbesii* WQ (156,975 bp). They all contained 84 protein-coding genes, 37 tRNAs, and 8 rRNAs. The hypervariable regions between three cp genomes were *atp*F-*atp*H, *pet*D, and *rps*15-*ycf*1. Phylogenetic analysis showed that *H. forbesii* LQ and *H. forbesii* WQ were closely related, followed by *H. forbesii* QX. This study showed that the three varieties of *H. forbesii* could be identified by the complete cp genome and specific DNA barcode (*trn*C-GCA-*pet*N) and provided a new idea for germplasm identification of similar cultivated varieties.

## Introduction

*Hansenia forbesii* H.Boissieu is a rare and endangered perennial herb belonging to the genus *Hansenia* (Apiaceae), and the Chinese name is "Kuan-ye-qiang-huo". The main chemical components isolated from *H. forbesii* are volatile oils, terpenoids, coumarins, sugars and glycosides, alkynes, phenolic acids, steroids, flavonoids, and alkaloids. It has anti-inflammatory, antibacterial, antioxidant, antiviral, anticancer, antipyretic, and analgesic activities, and has a significant impact on the cardiovascular and cerebrovascular, digestive, respiratory, and central nervous systems [1–4]. In the traditional production areas of *Hansenia*, the loss of germplasm resources is an extremely serious problem. As the problem of artificial breeding has not been solved, it is difficult to collect enough seeds, roots, and other reproductive materials in the genuine production areas and even enough genetic standard materials, showing that the severity of the risk of endangerment of *Hansenia* cannot be ignored. The wild-growing to domestic cultivation for *Hansenia* and the establishment of a standardized production base are the most effective ways to ensure the quality of medicinal materials, meet the medicinal requirements, protect wild resources and achieve sustainable development. Due to the wide distribution range and low altitude of *H. forbesii*, the requirements for environmental and soil conditions are relatively loose, the root system is well developed and the medicinal value is the same as that of *H. weberbaueriana*. Therefore, vigorously promoting the artificial cultivation of *H. forbesii* can reduce the pressure on this resource.

**Data Availability Statement:** The genome sequence data that support the findings of this study are openly available in GenBank of NCBI at

[https://www.ncbi.nlm.nih.gov] under accession no. ON 208134, ON208135, ON208136.

**Funding:** the Ministry of Finance and Ministry of Agriculture and Rural Areas: special subsidy of national modern agricultural industrial technology system (cars-21); Gansu Dingxi Science and Technology Project (dx2018n04). The sponsor is responsible for publishing decisions and guiding manuscript revisions in this study.

**Competing interests:** We declare that there is no conflict of interest regarding the publication of this article.

The main distribution types of *H. forbesii* are purple stem type, green stem type, large leaf type, small leaf type, root single branch type, root dispersed type, and other mixed species. The excellent varieties of *H. forbesii* have not been popularised and applied in production and cultivation, which cannot meet the needs of its industrial development. Moreover, there is a serious problem of mixed varieties in field production, which has a significant impact on its high-yield cultivation and is the bottleneck for the development of the *H. forbesii* industry. Before this study, we tried to address these issues by the following actions. Taking the wild *H. forbesii* (Wei-Qiang) resources in Gansu Province as the original materials, using the single plant optimization method and taking the plant agronomic characters, susceptibility, and drug grade as the indicators, we have selected two new varieties of *H. forbesii* (Long-Qiang and Qiang-Xuan) with high quality and high yield through line identification, line comparison, multi-point test, and production test. But it is not clear what kind of variation exists at the gene level and what is their phylogenetic relationship.

Chloroplast is an essential organelle in plant cells and it plays an important role in photosynthesis, carbon sequestration, and amino acid synthesis. Its complete genome contains a large amount of genetic information and its structure is highly conserved, including gene content, gene organization, and intron content within genes [5]. Chloroplast (cp) genome has also undergone intron loss, gene loss, gene duplication, gene rearrangement, pseudogenization, as well as uneven contraction and expansion of IRs regions. These genomic events can in turn become synapomorphies for entire clades [6]. In addition, many mutational events take place within the chloroplast genome including substitutions, insertions-deletions, inversions, microstructural changes, and oligonucleotide repeats [7–9]. The cp genome is slow evolving [10] and inherited from a single parent: maternally in most angiosperms [11] and paternally in some gymnosperms [12]. It is widely used in the study of plant molecular evolution and phylogeny, as well as in genetic transformation, genetic engineering, and molecular breeding [13–15]. Recently, with the gradual popularization and low cost of high-throughput sequencing technology, some scientists have proposed that cp genome data can be used for species identification and interspecific differentiation research [16–20]. There is no significant difference in some appearance indicators among the three *H. forbesii*. Through comparative analysis of their cp genome, the differences between them can be identified at the genetic level, and polymorphism markers can be developed to achieve variety identification. This is necessary for solving the problem of mixed varieties of *H. forbesii*, and has good significance for promoting excellent variety.

To find the gene hypervariable regions of three varieties of *H. forbesii* and determine their phylogenetic relationship, we used Illumina hiseq technology to sequence the cp genome. And we compared the characteristics of the cp genome, screened the different nucleotide sequences, and deciphered the genetic relationship to provide the basis for the differentiation of target varieties with excellent and optimal shapes.

## Materials and methods

### Materials and pretreatment

The fresh, tender, and healthy leaves of three *H. forbesii* were collected from the germplasm resource garden in Taoyang Town, Lintao County, Gansu Province in China on 30 September 2021. Professor Zhirong Sun, a scientist in the national traditional Chinese medicine system, identified that all the samples were *Hansenia forbesii* H.Boissieu (Apiaceae). The plant materials produced and used in this study comply with China guidelines and legislation. All the experiments were carried out to national and international guidelines. The aboveground forms of three *H. forbesii*, are shown in Fig 1 and described in S1 Table, which are named QX,

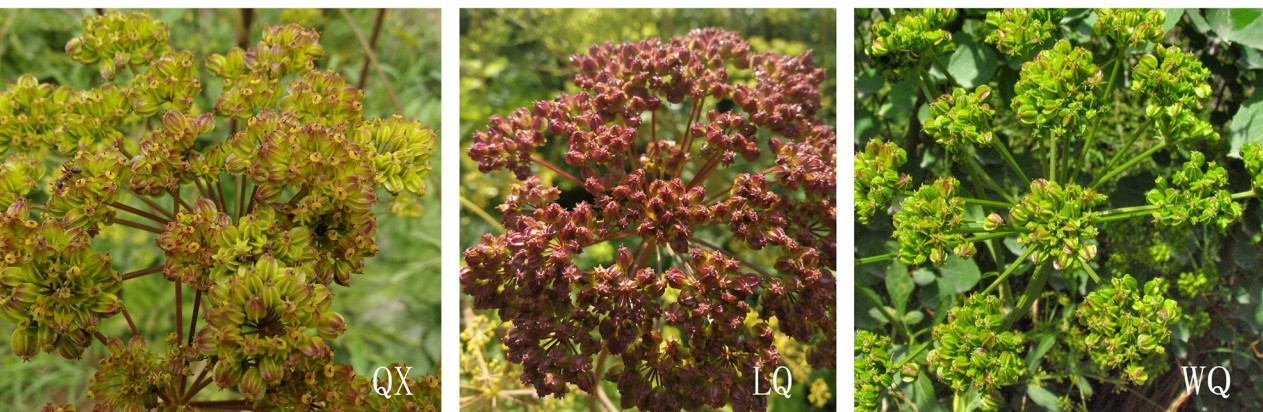

**Fig 1. The aboveground forms of three *H. forbesii*.**

LQ, and WQ according to their morphological characteristics and breeding process. After cutting off the veins of leaves with clean scissors, we put the remaining leaves into molecular bags, added silica gel, and saved them. The voucher specimens (QH210930) of all materials were kept in the herbal medicine library of the School of Chinese Materia Medica, Beijing University of Chinese Medicine.

## Total DNA extraction and library preparation

The total DNA was extracted using a small plant DNA Extraction Kit (Omega Bio-Tek, USA). Then we randomly broke qualified DNA samples using an ultrasonic crusher (Covaris, USA), and then completed the whole library preparation through the steps of end repair, adding A-tailing, adding sequencing connector, purification, and PCR amplification [21]. The qualified library was sequenced using the Illumina high-throughput sequencing platform (Illumina, NovaSeq 6000, USA).

## Chloroplast genome assembly and annotation

The high-quality reads were assembled using Get Organelle v.4.0 and then annotated by CpGAVAS2 [22]. The annotations of tRNA genes were confirmed by using tRNAscan-SE v.2.03 [23].

The cp genome of LQ, QX, and WQ were submitted to GenBank at the National Center of Biotechnology Information (NCBI), and the accession numbers were ON 208134, ON208135, and ON208136, respectively. Fully annotated plastome circular diagrams were drawn by OrganellarGenomeDRAW (OGDRAW) [24, 25].

## Codon usage

The protein-coding genes were extracted by Phylosuite v.1.2.2 [26]. Relative synonymous codon usage (RSCU) and codon usage values were analyzed by CodonW v.1.4.2. Moreover, the RSCU values were shown in a heatmap by TBtools [27].

## Repeat analysis and comparative analyses

Repetitive sequence analyses were performed using CPGAVAS2 analysis. Tandem repeats were identified using default settings by Tandem Repeats Finder [28]. The Misa. pl was used to screen the simple sequence repeats (SSRs) [29]. The scattered repetitive sequences were found

by using VMATCH. The REPuter was used to determine the size and location of the oligonu-cleotide repeats (ORs) [30]. The complete cp genome of three cultivated varieties was com-pared by mVISTA [31], and the genome of *H. forbesii* (NC035054) was used as the reference sequence for annotation. Sliding window analysis was conducted to assess the nucleotide diversity (Pi) values of the cp genome by DnaSP v6 (window length = 300 bp, step size = 25 bp). IRscope [32] was used to analyze inverted repeated traction and expansion at cp genome junctions.

### Identification and validation of barcode for varieties discrimination

According to the results of DNAsp, we chose the high variation region to distinguish the three varieties. Primers to discriminate between the three varieties under study were designed on the variable intergenic regions using Snapgene 6.2.1 (Snapgene from Insightful Science, available at http://www.snapgene.com, last used in 2023). PCR amplifications were performed in a final volume of 20 μL with 10 μL 2×Taq PCR Master Mix, 0.5 μM of each primer, 5 μL template DNA, and 4 μL ddH$_2$O following the manufacturer's instructions (Mei5 Biotechnology, Co., Ltd). All amplifications were carried out in a Pro-Flex PCR system (Applied Biosystems, Waltham, MA, USA) under the following conditions: denaturation at 95 ˚C for 3 min, followed by 36 cycles of 94 ˚C for 25 s and 55 ˚C for 10 s, and 72 ˚C for 2 min as the final extension following the manufacturer's instructions (Mei5 Biotechnology, Co., Ltd). PCR amplicons were visualized on 1% agarose gels, purified, and then subjected to bidirectional Sanger sequencing on an ABI 3730 XL instrument (Applied Biosystems, USA) using the same set of primers used for PCR amplification with BigDye v3.1 chemistry (Applied Biosystems) following manufacturer's instructions. All amplifications were repeated three for each variety.

### Phylogenetic analysis

Phylogenetic analysis was performed based on 37 complete cp genomes, including the three assembled sequences in our study, 34 cp genomes (30 Genus, 2 Family) downloaded from the NCBI and *Cornus officinalis* (NC042746) as outgroup. The shared protein-coding genes were extracted using Phylosuite v.1.2.2 software, and the MAFFT v.7.307 was used for align-ment [33]. Subsequently, the alignment was conducted using the GTR+I+Gevolution model based on Bayesian inference (BI) in MrBayes [34]. The parameter was set to run for five mil-lion generations and sampled every 1,000 generations, with all other settings left at their defaults, and the first 25% of each run was discarded as burn-in. The alignment was also eval-uated using bootstrap analysis on 1000 in a maximum likelihood (ML) by RAxML [35], with parameters: raxmlHPC-PTHREADS-SSE3 -fa -N 1000-m GTRGAMMA -x 551,314,260 -p 551,314,260 -o *Cornus_officinalis* _NC_042746-T 20, 1000 replications and best-fit model selection.

## Results

### Chloroplast genome information

The basic information of the cp genome of three *H. forbesii* is shown in Table 1 and compared with other species of the *Hansenia* genus and *Arabidopsis thaliana* (S2 Table). The length of the cp genome of *H. forbesii* QX was the longest in three varieties (157,181 bp), the total GC content of three cp genomes was about 37.6%, and they all had a typical four-region structure, in which the LSC (large single-copy region) was 86,167 bp (LQ), 86,361 bp (QX), and 86,188 bp (WQ), the SSC (small single-copy region) was 17,755 bp (LQ), 17,786 bp (QX) and 17,755

**Table 1. Basic information of cp genome of three varieties of *H. forbesii*.**

| Name | Overall length | LSC | SSC | IR | GC content (%) | protein-coding genes | tRNA genes | rRNA genes |
|------|---------------|-----|-----|-----|----------------|---------------------|-----------|-----------|
| *H.f.* LQ | 156,954 | 86,167 | 17,755 | 26,516 | 37.68 | 84 | 37 | 8 |
| *H.f.* QX | 157,181 | 86,361 | 17,786 | 26,517 | 37.66 | 84 | 37 | 8 |
| *H.f.* WQ | 156,975 | 86,188 | 17,755 | 26,516 | 37.68 | 84 | 37 | 8 |

GC: Percentage of G/C base number in total base number.

bp (WQ) and the IR (inverted repeats, including IRa and IRb) was 26,516 bp (LQ), 26,517 bp (QX), and 26,516 bp (WQ) (Fig 2). A total of 129 genes were found in these varieties, including 84 protein-coding genes, 37 tRNA genes, and 8 rRNA genes (Table 2). The number and types of introns were similar among the three *H. forbesii*. The four types of cp genes contained 45 photosynthesis-related genes, 29 self-replication-related genes, six other genes, and four unknown genes. Amongst them, *ndh*B, *rpl*2, *rpl*23, *rps*12, *rps*7, and *ycf*2 appeared twice. Eleven genes each contained one intron, including *rpl*2 (×2) and *ndh*B (×2), which were located in the IR, and the genes (*rps*16, *atp*F, *rpo*C1, *pet*B, *pet*D, and *rpl*16) were located in the LSC, and the *ndh*A was the only present in the SSC region. In addition, the *ycf*3 and *clp*P comprise two introns.

## Comparison of chloroplast introns and exons

By analyzing the cp genome annotation files of the three varieties of *H. forbesii*, we found that the introns and exons of 21 genes in the cp genome of the three varieties were different. QX had one more bp than LQ and WQ in terms of the introns in the *trn*K-UUU, *rps*16, *atp*F, *ycf*3, *clp*P, and *rpl*16, but one less bp in *rpo*C1 (S3 Table). Studies have shown that introns can reduce gene splicing and translation, slow down cell metabolism and reduce energy consumption [36] and the *Clp*P gene is also related to plant disease resistance, which may be the reason why QX has better disease resistance than LQ and WQ.

## Codon usage

The cp genome of three *H. forbesii* contained 64 codons encoding 20 amino acids. The result of the RSCU revealed that 32 codons were used frequently in these cultivars, with the highest frequency of UUA followed by AGA (Fig 3). Moreover, the codon exhibited a strong bias toward an A or T at the third position. The codons that contain A/T at the 3' end mostly have RSCU ≥1, whereas the codons are having C or G at the 3' end mostly have RSCU ≤1. Amino acid frequency analyses revealed the highest frequency of Leucine, whereas Tryptophane was a rare amino acid. In general, we found high similarities in codon usage and amino acid frequency among the three cultivated varieties, and both contain high AT content.

## Repeat analysis

Our analyses identified SSRs per genome composed of mono- to di- nucleotide repeating units (Fig 4a). The number and type of SSRs in LQ and WQ were similar, with 34 single nucleotide repeats and 2 dinucleotide repeats. QX contains 39 single nucleotide repeats more than LQ and WQ. Moreover, the main type of mononucleotide repeats was T. Oligonucleotide repeats in three varieties. Fig 4b showed that the number of repeats varied in three varieties. We discovered 49 repeats in three varieties (LQ: 26 forward and 23 palindromic, QX: 30 forward and 19 palindromic, and WQ: 26 forward and 23 palindromic). Most of the repeats ranged in size from

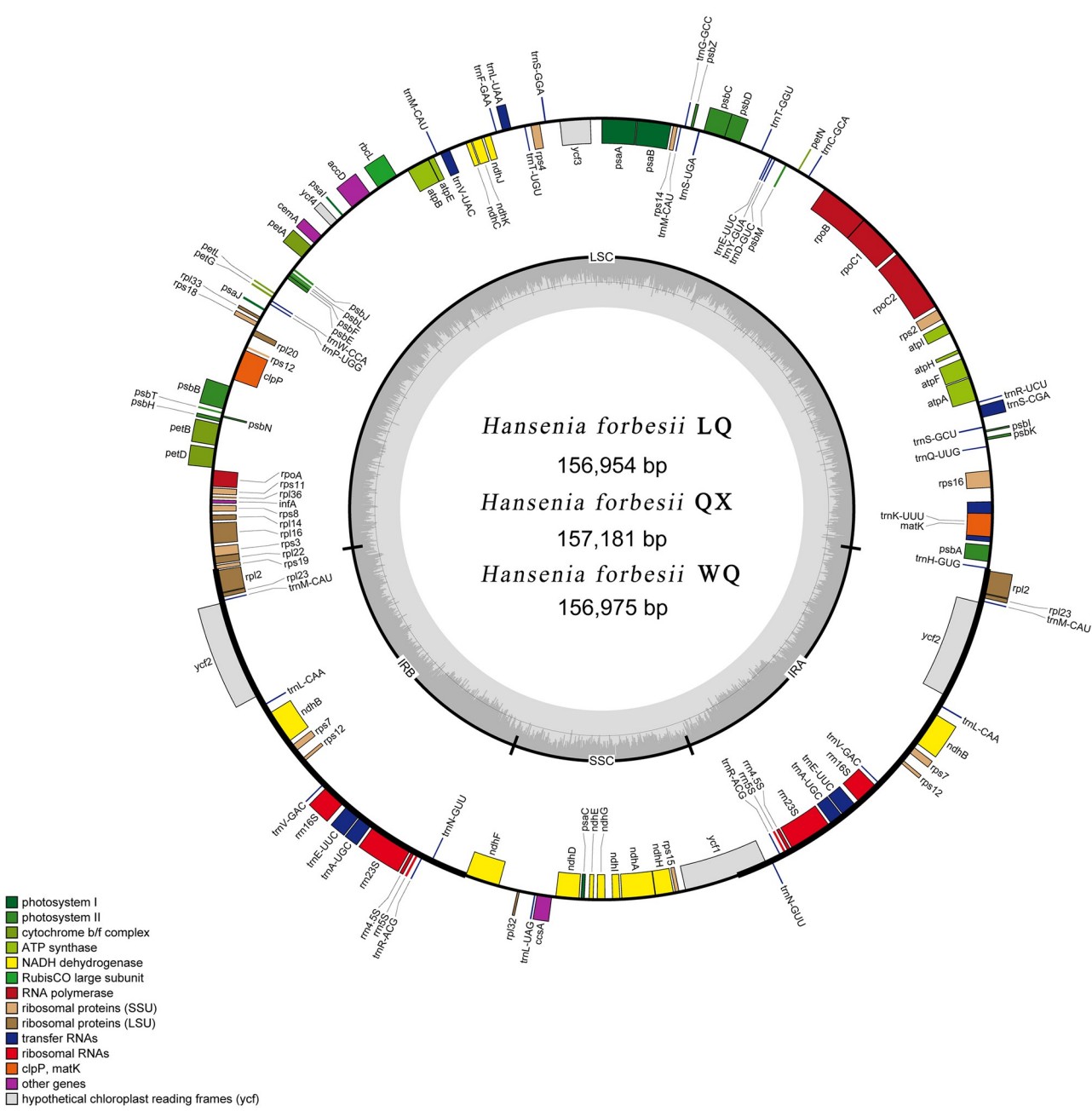

**Fig 2. Cp genome map of three varieties of *H. forbesii*.** Genes lying outside the circle are transcribed in the counterclockwise direction, while those insides are transcribed in the clockwise direction. The colored bars indicate different functional groups. The darker grey area in the inner circle denotes GC content, while the lighter grey corresponds to the AT content of the genome.

30 to 40 bp. This result showed that QX was more different than LQ and WQ. We also evaluated the number of repeats about the species' phylogenetic position using the topology in Fig 9.

## Inverted repeats contraction and expansion

The inverted repeats contraction and expansion revealed variation at LSC/IRs/SSC junctions (Fig 5). In three varieties, a truncated copy of the *rps*19 gene was found at the IRb/LSC

**Table 2. List of genes in the cp genome of three varieties of *H. forbesii*.**

| Category of genes | Group of genes | Name of genes | Number |
|---|---|---|---|
| rRNA | rRNA | *rrn*16S (×2), *rrn*23S (×2), *rrn*5S (×2), *rrn*4.5S (×2) | 8 |
| tRNA | tRNA | *trn*Y-GUA, *trn*W-CCA, \**trn*V-UAC, *trn*V-GAC(×2), *trn*T-UGU, *trn*T-GGU(×2), *trn*S-UGA, *trn*S-GGA, *trn*S-GCU, \**trn*S-CGA, *trn*R-UCU, *trn*R-ACG(×2), *trn*Q-UUG, *trn*P-UGG, *trn*N-GUU (×2), *trn*M-CAU(×4), *trn*L-UAG,\* *trn*L-UAA, *trn*L-CAA(×2),\* *trn*K-UUU, *trn*H-GUG, *trn*G-GCC, *trn*F-GAA, \**trn*E-UUC(×2), *trn*D-GUC, *trn*C-GCA,\**trn*A-UGC(×2) | 37 |
| Genes for photosynthesis | Subunits of ATP synthase | *atp*A, *atp*B, *atp*E, \**atp*F, *atp*H, *atp*I | 6 |
| | Subunits of photosystem II | *psb*A, *psb*B, *psb*C, *psb*D, *psb*E, *psb*F, *psb*I, *psb*J, *psb*K, *psb*L, *psb*M, *psb*N, *psb*T, *psb*Z, \*\**ycf*3 | 15 |
| | Subunits of NADH-dehydrogenase | \**ndh*A, \**ndh*B (×2), *ndh*C, *ndh*D, *ndh*E, *ndh*F, *ndh*G, *ndh*H, *ndh*I, *ndh*J, *ndh*K | 12 |
| | Subunits of cytochrome b/f complex | *pet*A, \**pet*B, \**pet*D, *pet*G, *pet*L, *pet*N | 6 |
| | Subunits of photosystem I | *psa*A, *psa*B, *psa*C, *psa*I, *psa*J | 5 |
| | Subunit of rubisco | *rbc*L | 1 |
| Self replication | Large subunit of ribosome | *rpl*14, \**rpl*16, \**rpl*2(×2), *rpl*20, *rpl*22, *rpl*23(×2), *rpl*32, *rpl*33, *rpl*36 | 11 |
| | DNA dependent RNA polymerase | *rpo*A, *rpo*B, \**rpo*C1, *rpo*C2 | 4 |
| | Small subunit of ribosome | *rps*11, *rps*12(×2), *rps*14, *rps*15, \**rps*16, *rps*18, *rps*19, *rps*2, *rps*3, *rps*4, *rps*7(×2), *rps*8 | 14 |
| Other genes | Subunit of Acetyl-CoA-carboxylase | *acc*D | 1 |
| | c-type cytochrom synthesis gene | *ccs*A | 1 |
| | Envelop membrane protein | *cem*A | 1 |
| | Protease | \*\**clp*P | 1 |
| | Translational initiation factor | *inf*A | 1 |
| | Maturase | *mat*K | 1 |
| Unkown | Conserved open reading frames | *ycf*1, *ycf*2 (×2), *ycf*4 | 4 |

\*Gene with one intron,

\*\*Gene with two introns.

junction; the *rpl*22 gene was found entirely in the LSC region; and the *rpl*2 gene was found entirely in the IRb region. Another truncated copy of the *ndh*F gene was found at the junction of IRb/SSC in all species, which starts in IRb regions and integrates into the SSC region. Moreover, a truncated copy of *ycf*1 was found in the SSC/IRa junction, which was longer in IRa of *H. forbesii*. In three varieties, *trn*H completely exists in LSC and only 3 bp from the junction of IRa/LSC. These results showed that the chloroplast genome of these three varieties did not expand or contract.

## Comparative cp genomic analysis

The cp genome of the three varieties was compared by mVISTA, and the *H. forbesii* (NC035054) was used as the reference sequence for annotation. The three varieties exhibited similar variation sites and degrees of variation (Fig 6). The coding regions (CDS) were more conserved than the intergenic spacers (IGS). The high divergence area in IGS was found in *pet*N-*psb*M, *trn*Y(GUA)-*trn*T(GGU), *ndh*C-*trn*V(UAC), *pet*A-*psb*J, *rps*8-*rpl*14, *rrn*4.5-*rrn*5. Furthermore, some mutations of CDS were found in *rpo*C1 and *rpl*2. Moreover, the result showed that IR regions had lower sequence divergence than LSC and SSC regions.

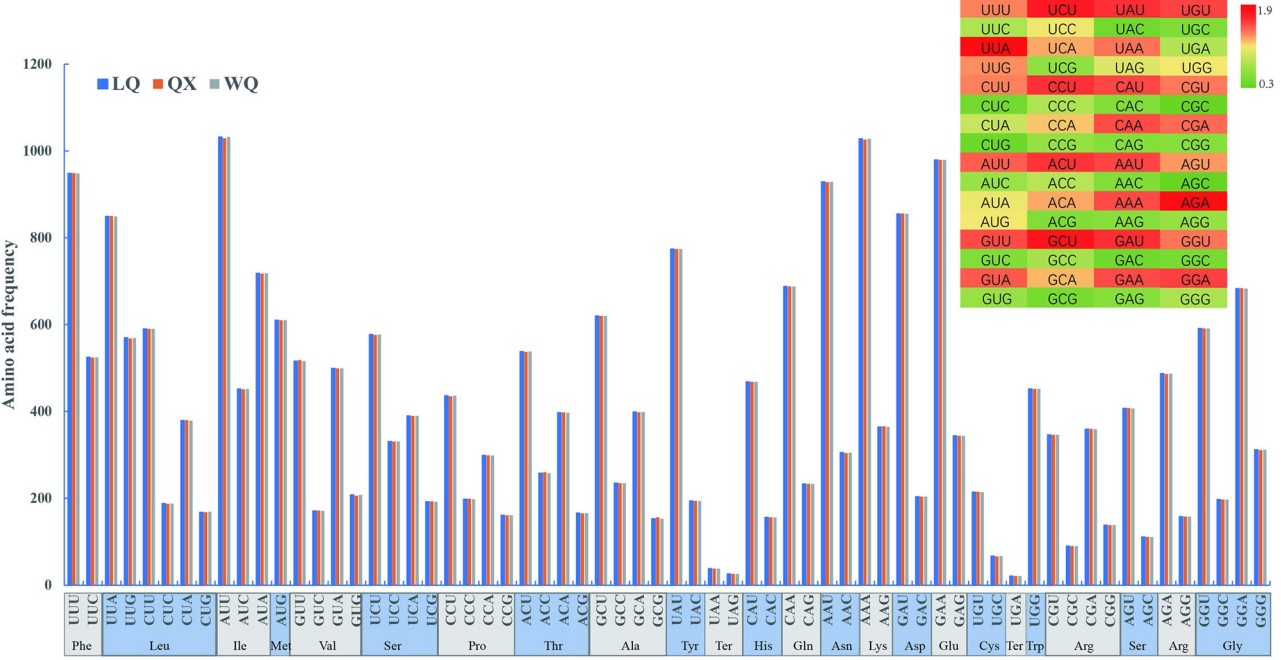

**Fig 3. The RSCU values of all protein-coding genes for three varieties.** Color key: the red values indicate higher RSCU values and the green values indicate lower RSCU values.

We used the DnaSP software to compare the nucleotide variation values (Pi) between the whole cp genome of the three varieties. The hypervariable regions were detected, and the sequence differences were analyzed. Sliding window analysis revealed that the nucleotide diversity values varied from 0 to 0.008. Three mutational hotspots in the LSC and SSC regions were identified, including *atp*F-*atp*H, *pet*D, and *rps*15-*ycf*1 (Fig 7). These can be used as potential sites for studying population genetics and the identification of these three varieties.

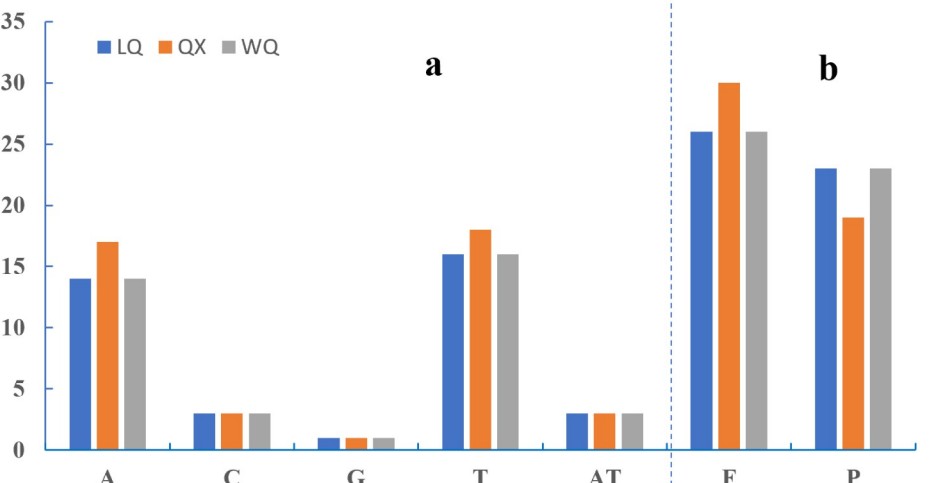

**Fig 4. Comparison of repeats in three varieties.** (a). SSR distributed situation in the cp genome of five species. (b). Long repeats classification of five species. P-palindromic repeats; F- forward repeats.

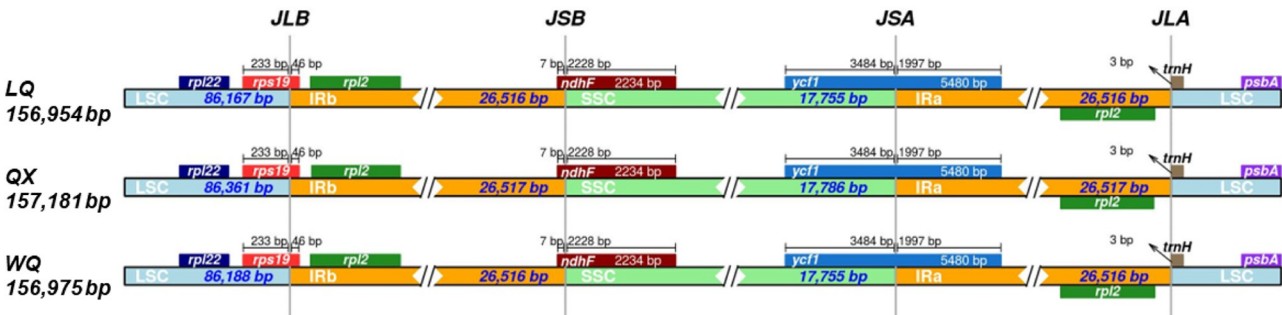

**Fig 5. Comparison of quadripartite junction sites in three varieties cp genome.** Gene transcribed clockwise are presented below the track, whereas transcribed counterclockwise are presented on top of the track. The start and end of each gene from the junctions have been shown with arrows. The T scale bar above or below the track shows genes integrated from one region of the cp to another. JLB (IRb /LSC), JSA (SSC/IRa), JSB (IRb/SSC), and JLA (IRa/LSC) denote the junction sites between the quadripartite regions of the genome.

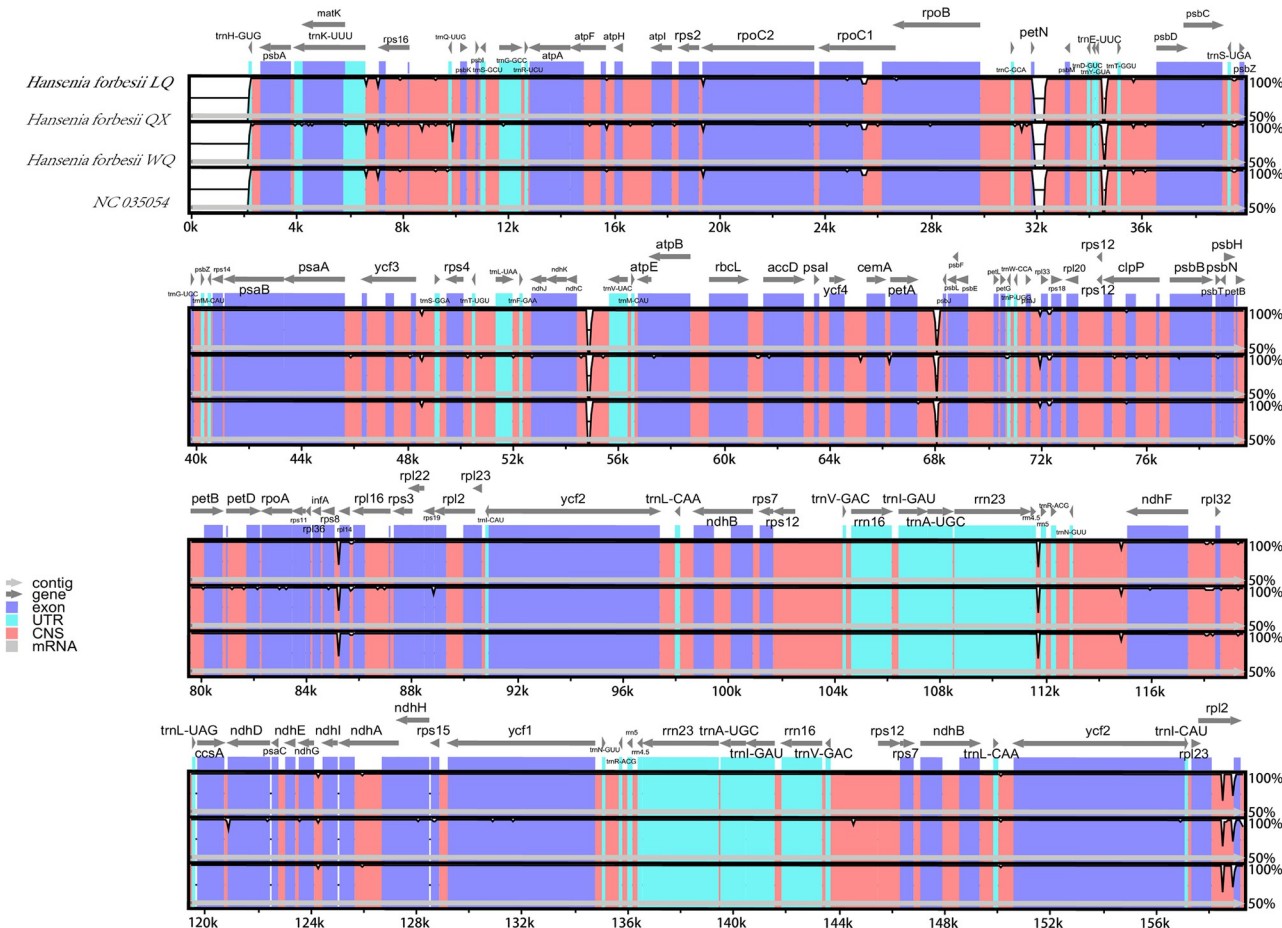

**Fig 6. Comparison of three varieties of cp genome using *H. forbesii* (NC035054) annotation as a reference.** The vertical scale indicates the percentage of identity, ranging from 50 to 100%. The horizontal axis shows the coordinates within the cp genome. Genome regions are color-coded as exons, introns, and intergenic spacer (IGS), and the Gray arrows indicate the direction of transcription of each gene. Annotated genes are displayed along the top.

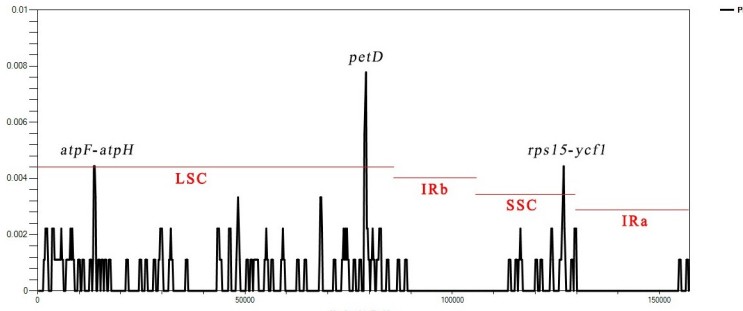

**Fig 7. Sliding window test of nucleotide diversity (Pi) in the multiple alignments of the three *H. forbesii* cp genomes.** Peak regions with a Pi value of>0.004 were labeled with loci tags of the genic or intergenic region names.

## Specific DNA barcode maker design for three varieties

To discriminate the three varieties, we selected two hypervariable regions, *trn*C-GCA—*pet*N (29,207–29,893 bp) and *trn*Q-UUG—*psb*K (7,894–8,266 bp), to develop specific DNA barcode. The related primer sequence is shown in S4 Table. PCR amplification of total DNA from three varieties resulted in products having the expected size (S1 Fig). In terms of the amplification of primer 2, QX exhibited more polymorphic loci compared to LQ and WQ. Under the amplification of primer 1, all three varieties showed a single bright band. The DNA fragments were extracted from each band and then subjected to Sanger sequencing. The sequencing results were identical to the expected sequences (Fig 8). The barcode has two specific Indel loci. These two variable loci can be used to differentiate three varieties.

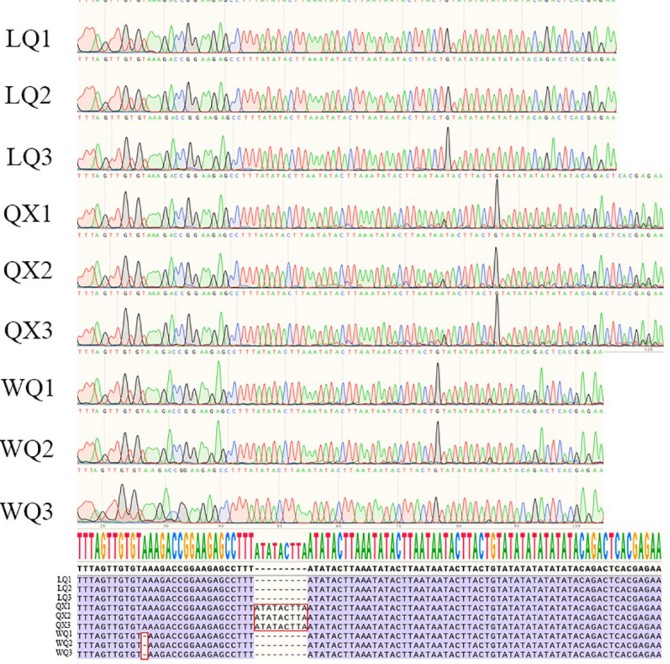

**Fig 8. Sequencing chromatograms of the barcode regions from three varieties with consensus sequence and alignment.**

## Phylogenetic analysis

The phylogenetic trees (ML and BI) were constructed by the shared protein-coding genes to investigate the phylogeny of the three *H. forbesii*, and *C. officinalis* (NC042746) was the outgroup. Most nodes were supported with maximum support (100% bootstrap support) on the consensus trees, and the topological structure of the two phylogenetic trees had perfect consistency (Fig 9). Three varieties of *H. forbesii* clustered with *Hansenia forbesii* (NC035054) and then merged with *H. forrestii* (NC035056) and *H. oviformis* (NC035055), indicating that LQ and WQ were closely related, followed by QX, and they were closely related to *Hansenia forbesii* (NC035054). Finally, they became a clade with *Changium smyrnioides* (NC053938), *Tongoloa silaifolia* (NC049062), *H. weberbaueriana* (NC035053), *Haplosphaera himalayensis* (NC056096). This clade had a close genetic relationship, in which QX was relatively distant from LQ and WQ.

## Discussion and conclusion

This study found that the three varieties of *Hansenia forbesii* had similar genetic relationships. Only the length of introns of some *H. forbesii* QX genes was longer than *H. forbesii* LQ and *H.*

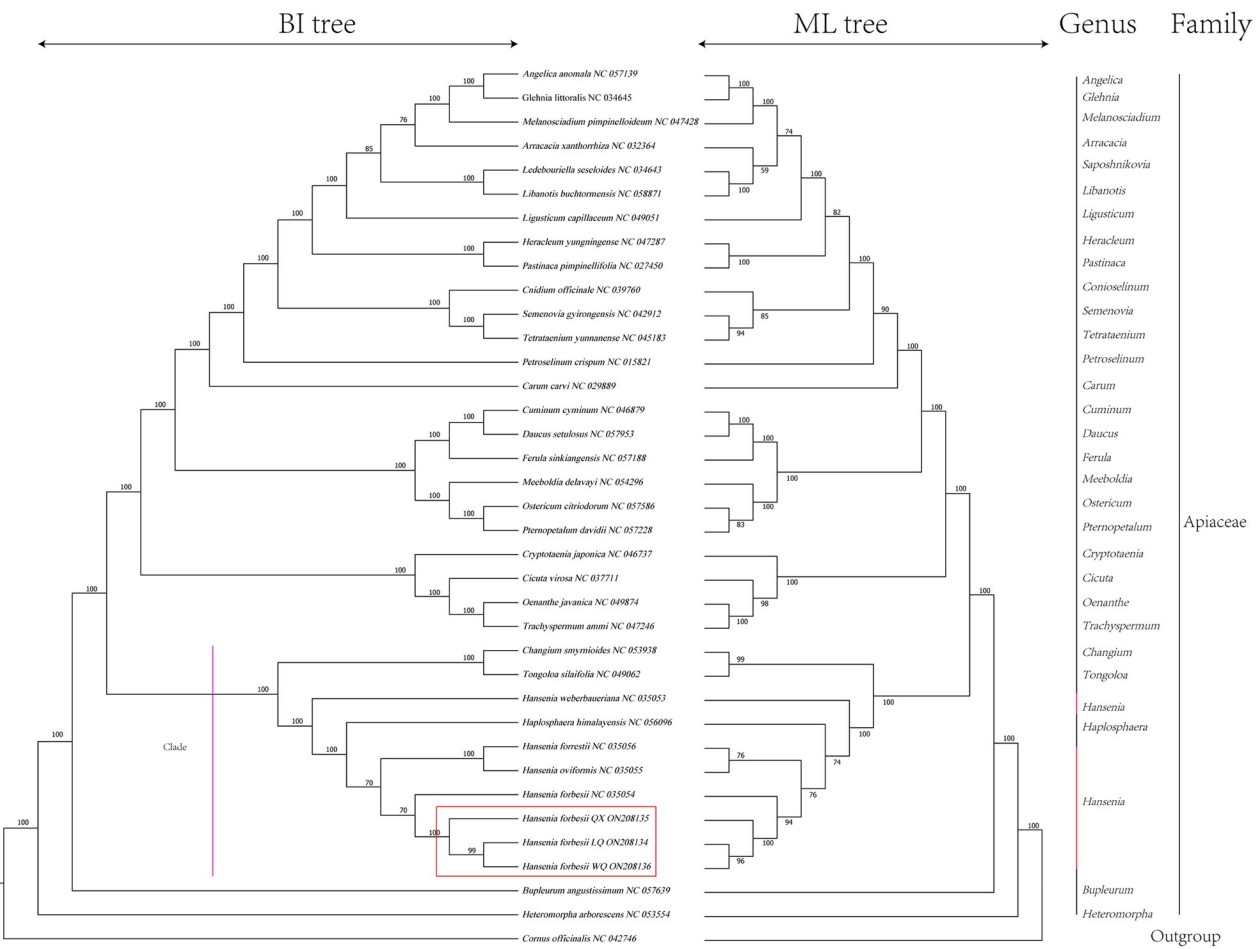

**Fig 9. BI and ML phylogenetic tree reconstruction containing the cp genome of 37 plants.** The number on the branch indicates that the support rate of the ML tree is greater than 50%; The *C. officinalis* were set as the outgroup.

*forbesii* WQ. We found three hypervariable regions between the cp genome of the three varieties of *H. forbesii*, including *atp*F-*atp*H, *pet*D, and *rps*15-*ycf*1. So the character differences between the three varieties of *H. forbesii* could be judged by the complete cp genome. Studies have shown that introns can reduce gene splicing and translation, slow down cell metabolism and reduce energy consumption [36]. QX had one more bp than LQ and WQ in terms of the introns in the *trn*K-UUU, *rps*16, *atp*F, *ycf*3, *clp*P, and *rpl*16, but one less bp in *rpo*C1. Taking Caseinolytic protease (*Clp*P) as an example, an ATP-dependent and highly conserved serine protease [37], it is essential for the regulation of the cellular stress response, cell homeostasis, and bacterial virulence [38, 39]. When encoding *Clp*P, the increase in introns length may affect gene splicing and translation, thereby affecting the expression of *Clp*P in QX, resulting in better disease resistance. Amino acid frequency analyses revealed the highest frequency of Leucine, Leucine in plants can enhance resistance to salt stress, enhance pollen vitality and germination, and enhance fragrance. The highest frequency of Leucine may also be one of the reasons why the three varieties have aromatic aromas.

The IR regions exhibited lower sequence divergence than LSC and SSC regions and were consistent with the previous comparisons of the cp genome [40]. Intergenic spacer regions are the most frequently used cp markers for phylogenetic studies at lower taxonomic levels in plants [41], as they are regarded as more variable and could provide more phylogenetically informative characters. In a previous study, *rps*15-*ycf*1 was identified as a mutational hotspot for identifying *Isodon rubescens* [18]. The *Pet*D located in the LSC region, as a protein-coding gene, contains an intron, which is consistent with previous genetic studies [40] and can also be used as a highly variable region for species identification. Phylogenetic analysis further showed that *H. forbesii* LQ and *H. forbesii* WQ were closely related, followed by *H. forbesii* QX, and they were closely related to *Hansenia forbesii* (NC035054), this also indirectly indicated the reliability of the data of three varieties of *H. forbesii*.

The Cp genome has a moderate rate of nucleotide evolution, which results in its suitability for species identification and phylogenetic studies at different taxonomic levels [42, 43]. Chen et al. identified the genus *Clerodendrum* with no significant difference in external morphology using the cp genome, they found two intergenic regions: *trn*H-GUG-*psb*A, *nhd*D-*psa*C, exhibiting a high degree of variations and showed that the complete cp genome can be used as a super-barcode to identify these *Clerodendrum* species [44]. Wu et al. compared five *Crataegus* cp genomes and found five highly variable regions, namely *pet*G-*trn*W-*CCA*, *trn*H-GUG-*psb*A, *trn*R-UCU-*atp*A, *pet*A-*psb*J, and *trn*T-GGU-*psb*D, indicating that the complete cp genome could also be used as a super-barcode to accurately authenticate the five Crataegus species [45]. Fu et al. obtained 48 cp genome from 16 *Taxus* species and *Pseudotaxus chienii* and constructed the ML evolutionary tree, which showed that the cp genome can successfully distinguish all *Taxus* species and can be used as a super barcode to identify species [46]. Lan et al. analyzed five cp genomes from *Artemisia* species, and showed that the cp genome can provide distinguishing features to help identify closely related *Artemisia* species and had the potential to serve as a universal super barcode for plant identification [20]. Most studies on the application of the cp genome have pointed out that a complete cp genome can be used as super barcodes for species identification. We found that the internal reason for the difference in the external phenotypes of the three varieties of *H. forbesii* can be reflected in the highly variable region of the cp genome, so the complete cp genome of three varieties of *H. forbesii* can be used as the DNA super barcode for identifying these three varieties. And *trn*C-GCA—*pet*N can be used as a specific DNA barcode to effectively identify the three varieties. This assists in solving the problem of mixed varieties in the actual production of *H. forbesii*, and can also be taken into account by other plants, which has great significance. In addition, the genetic relationship of the *H. forbesii* QX variety is far away from the other two varieties, so it can be

selected as the first-generation variety for single plant optimization and pure breeding to ensure its excellent phenotypes, and it can be popularized and applied on a large area. This study can provide a reference for the sustainable utilization of *Hansenia forbesii* resources and the identification of different varieties with different phenotypes of the same species.

## Supporting information

**S1 Table. The aboveground forms of three *H. forbesii*.**
(XLSX)

**S2 Table. Comparison of chloroplast genome basic information between three *H. forbesii* and the genus *Hansenia* and *Arabidopsis thaliana*.**
(XLSX)

**S3 Table. The genes with introns of the cp genome from three varieties and the length of exons and introns.**
(XLSX)

**S4 Table. The primers capturing hypervariable regions.**
(XLSX)

**S1 Fig. The gel electrophoresis results of the PCR products.** Lane M was the marker of DL 2000. The lanes from left to right corresponded to products of LQ1, LQ2, LQ3, QX1, QX2, QX3, WQ1, WQ2, and WQ3, respectively.
(TIF)

## Author Contributions

**Data curation:** Chenghao Zhu.

**Formal analysis:** Chenghao Zhu, Yuan Jiang, Yu Bai.

**Funding acquisition:** Shengjian Dong, Sun Zhirong.

**Methodology:** Yuan Jiang.

**Resources:** Chenghao Zhu, Yu Bai.

**Software:** Yuan Jiang.

**Writing – original draft:** Chenghao Zhu.

**Writing – review & editing:** Yuan Jiang, Shengjian Dong, Sun Zhirong.

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
