## [Decision Letter · Decision Letter 0]

20 Apr 2023

PONE-D-23-04874Comparative study on chloroplast genomes of three Hansenia forbesii varieties (Apiaceae)PLOS ONE

Dear Dr. zhirong,

Thank you for submitting your manuscript to PLOS ONE. After careful consideration, we feel that it has merit but does not fully meet PLOS ONE’s publication criteria as it currently stands. Therefore, we invite you to submit a revised version of the manuscript that addresses the points raised during the review process.

We look forward to receiving your revised manuscript.

Kind regards,

Shyamal Krishna Talukder, Ph.D.

Academic Editor

PLOS ONE

https://www.frontiersin.org/articles/10.3389/fpls.2022.1099856/full

https://www.sciencedirect.com/science/article/pii/S0888754319308420?via%3Dihub

In your revision ensure you cite all your sources (including your own works), and quote or rephrase any duplicated text outside the methods section. Further consideration is dependent on these concerns being addressed.

“the Ministry of Finance and Ministry of Agriculture and Rural Areas: special subsidy of national modern agricultural industrial technology system (cars-21); Gansu Dingxi Science and Technology Project (dx2018n04).”

“This research was supported by the Ministry of Finance and Ministry of Agriculture and Rural Areas: special subsidy of national modern agricultural industrial technology system (cars-21); Gansu Dingxi Science and Technology Project (dx2018n04).”

“the Ministry of Finance and Ministry of Agriculture and Rural Areas: special subsidy of national modern agricultural industrial technology system (cars-21); Gansu Dingxi Science and Technology Project (dx2018n04).”

Additional Editor Comments:

The manuscript is well written. Comparative analysis needs to be more detailed in terms of number of polymorphic loci (SNP, indels, deletion and duplication). The authors should report indels and large deletions and make an effort to design primers capturing deletions, indels and hypervariable regions, so that polymorphic markers can be developed for varietal discrimination. Discussion needs to be more precise in relation to functional changes in proteins.

Reviewers' comments:

Reviewer's Responses to Questions

**Comments to the Author**

1. Is the manuscript technically sound, and do the data support the conclusions?

Reviewer #1: Yes

Reviewer #2: Yes

2. Has the statistical analysis been performed appropriately and rigorously? 

Reviewer #1: Yes

Reviewer #2: No

3. Have the authors made all data underlying the findings in their manuscript fully available?

Reviewer #1: Yes

Reviewer #2: Yes

4. Is the manuscript presented in an intelligible fashion and written in standard English?

Reviewer #1: Yes

Reviewer #2: Yes

5. Review Comments to the Author

Reviewer #1: Discussion and Conclusion section I would like to request more precisely indicate actual fact compare with others findings and real application in the field of life science .

Discussion and Conclusion section I would like to request more precisely indicate actual fact compare with others findings and real application in the field of life science .

Reviewer #2: This is an interesting manuscript by Chenghao Zhu et al. elucidate the chloroplast genome of Hansenia forbesii varieties. The manuscript is well written and organized. Alike the interesting manuscript, I have found some minor issues that should be considered in your manuscript in future. Hence, the authors should take into consideration the following points in order to improve the manuscript.

1. Detailed descriptions of the chloroplast genome's usual properties should be contained in a single paragraph in the introduction. Moreover, the necessity of exploring the chloroplast genome in Hansenia forbesii is not clearly presented also.

2. A supplementary table showing the basic information of previously decoded chloroplast genome related to Hansenia species and model plant

3. Lines 134–136 gene type counting error.

4. In line 148-150 we know 3bp create a code, then in protein sequence one more or one less bp how create a three letter code.

5. Table 1: There is no statistical analysis in presenting the quantitative parameters.

6. How did you separate chloroplast DNA from the nuclear DNA?

7. How many biological replications were considered during library preparation for sequencing?

8. Line 93-94: National Center of Biotechnology Information. It must be National Center for Biotechnology Information.

9. The authors failed to interpret the results of three chloroplast genome differences, which resulted no molecular description of any morphological differences among the three Hansenia forbesii varieties.

6. PLOS authors have the option to publish the peer review history of their article (what does this mean?). If published, this will include your full peer review and any attached files.

Reviewer #1: No

Reviewer #2: **Yes: **Borhan Ahmed

---

## [Author Response · Author response to Decision Letter 0]

28 Apr 2023

27- April-2023

Ref. No.: PONE-D-23-04874

Title: Comparative study on chloroplast genomes of three Hansenia forbesii varieties (Apiaceae)

RESPONSE LETTER

Dear Editor:

We greatly appreciate the referees and your efforts in handling our manuscript (ID: PONE-D-23-04874). We thank you for your favorable consideration and the referees' insightful comments.

We have already carefully read the comments from the referees and revised the entire manuscript according to their suggestions. Besides, according to the reviewers' advice, most sentences have been rewritten, so some sentences and words do not exist. All of these changes have been marked in RED. And the whole article has been polished by a professional native speaker.

We are confident that the quality of the revised paper has dramatically improved, and we hope that the revised version is acceptable for publication in PLOS ONE.

The responses to the referees' comments are listed below point by point. Please feel free to contact us with any questions, and we are looking forward to receiving your response.

I thank you most sincerely for your time and consideration.

Best regards,

School of Chinese Materia Medica, Beijing University of Chinese Medicine

szrbucm67@163.com

Yangguang South Street, Fangshan District, 102488, China

Tel: +86 (010)84738334;

Editor and Reviewer comments:

1. Please ensure that your manuscript meets PLOS ONE's style requirements.

Response: Thanks for the editor's careful review. We have made modifications to the full text according to the format requirements of your journal, as shown in Red. In addition, we have reviewed the format again with reference to two recent articles in your journal. 

(1) PLoS ONE 18(3): e0277471. https://doi.org/10.1371/journal.pone.0277471

(2) PLoS ONE 18(2): e0277809. https://doi.org/10.1371/journal.pone.0277809

2. We noticed you have some minor occurrences of overlapping text with the following previous publication(s), which need to be addressed.

Response: Thank you for the editor's reminder. The sequencing methods for chloroplast genome are often similar and may result in text overlap. We have added citations to these two articles in the methods section of the revised manuscript and ensured that all sources are cited. Thanks!

Response: Thank you again for the editor's careful review. We do not require a work permit for our research. The collection of plant materials complies with relevant policies in China. The methods section has been explained in the revised manuscript, and the full name of the storage institution for the materials has also been explained. Thanks!

4. Please state what role the funders took in the study. If the funders had no role, please state: "The funders had no role in study design, data collection, and analysis, the decision to publish, or preparation of the manuscript."

Response: Thank you for the editor's review work. The sponsor is responsible for publishing decisions and guiding manuscript revisions in this study. We have added this amended Role of Funder statement in our cover letter. Please also edit and modify the online submission form. Thanks!

5. Please remove any funding-related text from the manuscript and let us know how you would like to update your Funding Statement. 

Response: Thank you for the editor's review work. We have removed funding-related text from the manuscript. And we have added this amended Role of Funder statement in our cover letter. Please also edit and modify the online submission form. Thanks!

6. Please include captions for your Supporting Information files at the end of your manuscript, and update any in-text citations to match accordingly.

Response: Thank you for the editor's review work again. We have added the title of the supporting information file at the end of the manuscript and updated the citation of the text. Please refer to it. Thanks!

7. Please review your reference list to ensure that it is complete and correct.

Response: Thank you to the editor for their rigorous review of each section of our manuscript. We have carefully reviewed the references and ensured their accuracy and completeness, and there are no articles that have been withdrawn. Thanks!

Editor: 

1. Comparative analysis needs to be more detailed in terms of number of polymorphic loci (SNP, indels, deletion and duplication). The authors should report indels and large deletions and make an effort to design primers capturing deletions, indels and hypervariable regions, so that polymorphic markers can be developed for varietal discrimination. Discussion needs to be more precise in relation to functional changes in proteins.

Response: Thank you for the professional comments. This provides us with a great suggestion. (1) We analyzed the high variability region again, then discovered specific barcodes, and performed sequencing validation. The additional captured primer 2 can amplify polymorphic sites to identify QX from LQ and WQ by PCR and gel electrophoresis. Through amplification and sequencing of primer 1, it is found that the primer can identify the three varieties separately, and adds comparative analysis of indel. (2) The content related to protein functional changes among the three varieties in this manuscript is only the inconsistent length of introns in some genes, which leads to protein functional changes. Due to the fact that introns can reduce gene splicing and translation, thereby affecting the expression of related proteins, discussions in this section have been increased, while also increasing the frequency analysis of amino acids encoded by the chloroplast genome. Thanks!

Reviewer #1: 

1.Discussion and Conclusion section I would like to request more precisely indicate actual fact compare with others findings and real application in the field of life science.

Response: We think this is an excellent suggestion. We have found that most studies on the application of chloroplast genomes in species identification have pointed out that complete genomes can be used as super barcodes for identification. Our research has found that specific barcodes can also be used for variety identification. This will make the identification work more accurate and effective. And in the field of life science, this has assisted in solving the problem of variety mixing in the actual production of Hansenia forbesii, and can also be borrowed from other plants, it has great significance. This section has been added to the discussion and conclusion. Thanks!

Reviewer #2:

1. Detailed descriptions of the chloroplast genome's usual properties should be contained in a single paragraph in the introduction. Moreover, the necessity of exploring the chloroplast genome in Hansenia forbesii is not clearly presented also.

Response: We agree with the referee's comments. We have provided a detailed description of the common properties of the chloroplast genome, and have clearly pointed out the necessity of exploring the chloroplast genome in the introduction. We have found molecular means to identify three Hansenia varieties and solve the problem of variety mixing in actual production. Thanks!

2. A supplementary table showing the basic information of previously decoded chloroplast genome related to Hansenia species and model plant. 

Response: We sincerely appreciate the valuable comments. We have supplemented the chloroplast genome information of the three Hansenia varieties in this study, as well as the Hansenia species and model plants reported in previous literature in S2 Table. Thanks!

3. Lines 134–136 gene type counting error.

Response: Please forgive our carelessness. We have corrected the error and reverified the number of gene types, including 29 self-replication related genes, six other genes, and four unknown genes. Thank you for your seriousness!

4. In line 148-150 we know 3bp create a code, then in protein sequence one more or one less bp how create a three letter code.

Response: Thank you for your suspicion. Due to a clerical error, what we want to express here is the intron of certain genes, which is either 1 bp more or 1 bp less. There may be ambiguity in the text, which has been revised. Thanks!

5.Table 1: There is no statistical analysis in presenting the quantitative parameters.

Response: Thank you for your suggestion. The original data presented the differences in the external morphology of the three varieties. After testing the relevant indicators of external traits, statistical analysis found that there was no significant difference, so there was no statistical indicator. As these external data did not have statistical differences, this study conducted a comparison between chloroplast genomes. The data has been placed in the S1 table. Thanks!

6. How did you separate chloroplast DNA from the nuclear DNA?

Response: Thank you for your question. Firstly, the total DNA of the sample is extracted during extraction, including nuclear, chloroplast, and mitochondrial DNA. During the sequencing process, total DNA was tested and a database was established. Then, the database was filtered using chloroplast genome data from all species on NCBI, and then assembled using the reported Hansenia forbesii (NC035054) as the template. After assembly, the chloroplast genomes of three varieties were obtained. Thanks!

7. How many biological replications were considered during library preparation for sequencing?

Response: Thank you for your question again. In chloroplast genome testing, there is 1 biological duplication (3 leaves on 1 plant), and in barcode validation, there are 3 biological replicates (3 plants per variety). Thanks!

8. Line 93-94: National Center of Biotechnology Information. It must be National Center for Biotechnology Information 

Response: Please excuse this clerical error. We have made corrections. Thanks!

9. The authors failed to interpret the results of three chloroplast genome differences, which resulted no molecular description of any morphological differences among the three Hansenia forbesii varieties.

Response: Thank you very much for providing this suggestion. We analyzed the high variability region, then discovered DNA barcodes, and performed sequencing validation. The reason for the differences between the three chloroplast genomes may be due to deletions, mutations, and duplications in some gene regions. Polymorphic sites were excavated and sequencing validation was conducted, and it was found that trnC-GCA-petN can be used as a specific DNA barcode to effectively identify the three varieties. This section has been added to the manuscript. Thanks!

---

## [Editor Report · Decision Letter 1]

19 May 2023

Comparative study on chloroplast genomes of three Hansenia forbesii varieties (Apiaceae)

PONE-D-23-04874R1

Dear Dr. Zhirong,

We’re pleased to inform you that your manuscript has been judged scientifically suitable for publication and will be formally accepted for publication once it meets all outstanding technical requirements.

Within one week, you’ll receive an e-mail detailing the required amendments. When these have been addressed, you’ll receive a formal acceptance letter, and your manuscript will be scheduled for publication.

Kind regards,

Shyamal Krishna Talukder, Ph.D.

Academic Editor

PLOS ONE

---

## [Editor Report · Acceptance letter]

23 May 2023

PONE-D-23-04874R1 

Comparative study on chloroplast genomes of three *Hansenia forbesii* varieties (Apiaceae) 

Dear Dr. zhirong:

I'm pleased to inform you that your manuscript has been deemed suitable for publication in PLOS ONE. Congratulations! Your manuscript is now with our production department. 

Kind regards, 

on behalf of

Dr. Shyamal Krishna Talukder 

Academic Editor

PLOS ONE